# Remodeling of the Neurovascular Unit Following Cerebral Ischemia and Hemorrhage

**DOI:** 10.3390/cells11182823

**Published:** 2022-09-09

**Authors:** Yoshimichi Sato, Jaime Falcone-Juengert, Teiji Tominaga, Hua Su, Jialing Liu

**Affiliations:** 1Department of Neurological Surgery, UCSF, San Francisco, CA 94158, USA; 2Department of Neurological Surgery, SFVAMC, San Francisco, CA 94158, USA; 3Department of Neurosurgery, Graduate School of Medicine, Tohoku University, 1-1 Seiryo-machi, Aoba-ku, Sendai 980-8574, Japan; 4Department of Anesthesia, UCSF, San Francisco, CA 94143, USA; 5Center for Cerebrovascular Research, UCSF, San Francisco, CA 94143, USA

**Keywords:** NVU, blood–brain barrier, pericyte, angiogenesis, arteriogenesis

## Abstract

Formulated as a group effort of the stroke community, the transforming concept of the neurovascular unit (NVU) depicts the structural and functional relationship between brain cells and the vascular structure. Composed of both neural and vascular elements, the NVU forms the blood–brain barrier that regulates cerebral blood flow to meet the oxygen demand of the brain in normal physiology and maintain brain homeostasis. Conversely, the dysregulation and dysfunction of the NVU is an essential pathological feature that underlies neurological disorders spanning from chronic neurodegeneration to acute cerebrovascular events such as ischemic stroke and cerebral hemorrhage, which were the focus of this review. We also discussed how common vascular risk factors of stroke predispose the NVU to pathological changes. We synthesized existing literature and first provided an overview of the basic structure and function of NVU, followed by knowledge of how these components remodel in response to ischemic stroke and brain hemorrhage. A greater understanding of the NVU dysfunction and remodeling will enable the design of targeted therapies and provide a valuable foundation for relevant research in this area.

## 1. Introduction

The Neurovascular Unit (NVU) is a novel and transforming concept formalized in 2001 by the Stroke Progress Review Group of the National Institute of Neurological Disorders and Stroke [1]. As implied by its name, the NVU is constituted by elements of the nervous and vascular systems. The neural elements consist of astrocytes, pericytes, microglia, peripheral immune cells, and neurons, while the vascular components include Endothelial Cells (ECs), Vascular Smooth Muscle Cells (VSMCs), and pericytes. Together, these cellular networks are responsible for the cellular interplay from neuron-to-vessel communication, Neurovascular Coupling (NVC), vessel-to-neuron signaling, vasculo-neuronal coupling to maintain brain homeostasis, and responding to inflammation and disease. The unique structure of the NVU along with transmembrane proteins forms a barrier that regulates the movement of molecules between the blood and brain, called the blood–brain barrier (BBB) (Figure 1). Many neurological diseases are associated with the breakdown of this interplay, resulting in increased permeability of the BBB and neuronal dysfunction [2,3]. In the event of a stroke, the NVU plays a pivotal role in the progression of stroke injury and remains the main target of neuroprotective therapy.

## 2. The Structural Components of the Neurovascular Unit (NVU)

### 2.1. Neurons

Different types of neurons including noradrenergic [4,5], serotonergic [6], cholinergic [7,8], and GABAergic [9] neurons have been shown to make distinct connections with other cells of the NVU, allowing regional brain activity to be metabolically coupled to blood flow [10]. Evidence shows that BBB opening may be a selective compensatory event rather than a simple anatomical disruption, implicating that interaction between neurons and the brain microvasculature may regulate both blood flow and BBB permeability.

### 2.2. Endothelial Cells

ECs line blood vessels in the brain and are joined with one another by specialized Tight Junctions (TJs) and Gap Junction (GJ) proteins consisting of various molecular components [11,12,13,14]. These TJs form the BBB, a physical barrier between blood and brain parenchyma [15], and regulate the permeability of the EC layer that keeps unwanted molecules such as toxins from entering the brain. This organization supports the current view that the BBB is not just a physical “barrier”, but a dynamic and metabolic interface [16,17].

Capillaries make up the microcirculation and are divided into three types: continuous, fenestrated, and sinusoidal. Brain capillaries belonging to the continuous type have no perforations on the vessel wall, allowing only small molecules to pass through. In contrast, fenestrated capillaries in the kidneys and GI tract along with sinusoidal capillaries in the lymphoid organs are leakier and have small and large pores, respectively. They allow for the passage of large molecules and even cells compared with the brain capillaries. By directly isolating tissue-specific mRNAs using the RiboTag transgenic mouse model [18], a recent study compared the transcriptome profiles of ECs from the brain, heart, and lung using RNA sequencing. The authors found that each EC type had a distinct genetic signature under normal conditions and an organ-specific response to lipopolysaccharide. Most surprisingly, ECs turned on genes that were expressed by the surrounding tissue upon stimulation, indicating organ-specific endothelial plasticity and adaptation [19]. Specifically in the brain, genes involved in processes akin to neuronal function such as synapse organization, neurotransmitter transport, axon development, and ion transport regulation are also enriched in the brain ECs [19].

Endothelial cells also interact with other components in the brain, such as pericytes, astrocytes, neurons, and the ECM to maintain the normal function of NVU by forming the BBB [16,20]. The interplay among partners in the NVU and the integrity of the BBB is key to brain homeostasis. The interaction between capillaries and neuronal components in neurovascular coupling regulates CBF through the detection of changes in shear stress of the endothelial wall and the supply of oxygen and necessary nutrients in the brain tissue. BBB breakdown leads to an influx of toxic molecules towards the brain, causing neuronal injury and neurodegenerative changes. While the NVU and BBB are critical for the maintenance of Central Nervous System (CNS) homeostasis, they could also interfere with the delivery of therapeutic drugs into the CNS via systemic administration for the treatment of neurological diseases. Further research and clinical trials are needed to successfully deliver necessary drugs to the CNS without causing inadvertent permeability of the BBB.

### 2.3. Pericytes

Pericytes surround cerebral vessels and are intimately in contact with the ECs embedded in the vascular basement membrane (BM) through gap junctional complexes around pericyte somata [21,22], called peg-socket contacts. Like endothelial cells, pericytes are attached to extracellular matrix proteins of the BM by different integrins that control junctional complex protein expression [23], affecting the functionality of the BBB by controlling both the structure of TJs and the rate of vesicular trafficking. They are also involved in various vascular functions such as BBB formation andmaintenance, angiogenesis, vessel maturation, regulation of blood flow, and immune cell trafficking [24,25,26,27]. Pericytes can inhibit the expression of genes that promote vessel permeability. Pericytes control neuroinflammation by reducing leukocyte trafficking in the regions of blood vessels they cover [24]. A study using pericyte-deficient mice showed a correlation between the reduction of tight junction and adherens junction proteins and the resulting increase in paracellular leakage and eventually BBB breakdown [25]. The resulting permeability leads to an influx of neurotoxic macromolecules, water, and larger molecules that are normally incapable via increased endothelial transcytosis [26,27], and a reduction in capillary blood flow due to microvascular degeneration [28].

Capillary vessels dilate to accommodate oxygen demand in the brain in response to physiological stimulation such as hypercapnia or sensory stimulation. Earlier studies implied that pericytes can contract or relax, resulting in changes in the capillary diameter under various physiological and pathological conditions [29,30,31,32]. To further distinguish which cells in the vessel zones respond to stimulation, a study using optogenetics found that it was SMCs rather than pericytes that contracted when mural cells expressing ChR2 were stimulated [30]. However, another study found that pharmacological inhibitor of myosin contraction signaling along with optical ablation of capillary pericytes resulted in consistent dilation of regions lacking pericyte contact, leading to aberrantly increased flux of blood flow in the uncovered capillary vessels [21,33]. A recent scRNAseq study suggests that pericytes do have the molecular machinery to regulate vessel diameter since capillary pericytes express receptors for vasoactive mediators including L-type voltage-gated calcium channels and those involved in smooth muscle cell actomyosin contraction [34], providing pivotal support for the modulatory role of pericytes in controlling capillary diameters and homogenizing blood flow to facilitate oxygen extraction, particularly during functional hyperemia. Apart from their modulatory role, pericytes seem multipotent and are known to differentiate into both neural and vascular lineage cells after brain ischemia [35,36].

The same group of investigators who found pericytes drive capillary vessel dilation reported that pericyte loss and capillary dilation caused by focal ablation of pericytes with laser was exacerbated in the aged brain, resulting in increased flow heterogeneity in capillary networks. Although the remodeling of neighboring pericytes restored endothelial coverage and vascular tone within days, this process was slower in the aged brain and led to persistent capillary dilation [37]. This suggests that pericytes do communicate with one another and work together to maintain neurovascular coupling. In support of this notion, one study showed that neighboring pericytes in the mouse retina can communicate with each other in response to light stimulation by forming interpericyte tunneling nanotubes that become a functional network with an open-ended proximal side and a closed-ended end-foot that connects with distal pericyte processes via gap junctions, serving as a conduit for intercellular Ca^2+^ waves between pericytes [38].

### 2.4. Astrocytes

Linking the vascular and neural systems are perivascular astrocytes, whose astrocytic processes almost completely encapsulate the abluminal EC surface of brain vessels. Astrocytes play a role in the development of junctional complexes and physically link neighboring neurons with blood vessels [39,40,41], allowing them to detect changes in the neuronal microenvironment and adjust the microvasculature appropriately [42,43]. Perivascular astrocytes increase the tightness of TJs [44], promote the expression of endothelial transporters [45], and enzymes associated with the endothelial barrier [40]. Through the secretion of bioactive substances, astrocytes provide physical support and strengthen the BBB, leading to TJ modulation. Gap junctions present between astrocytic end feet and vessel walls mediate intercellular communication and solute movement between them, such as water and ion exchange across the brain microvascular endothelium [40,46]. Water channel aquaporin 4 is noticeably expressed on astrocytic end feet and regulates water movement between the blood and brain [47]. Astrocytes are known for their roles in responding to CNS injury by taking up excess glutamate from the extracellular space and converting it to glutamine [48] to aid in the generation of new neurons, remodeling synapses, and recycling neurotransmitters [24]. Critical to neuronal survival and repair, a large part of this function is mediated by gap junction proteins that connect astrocyte networks into a functional syncytium [13,49,50,51]. Through the secretion of proinflammatory (Interleukin (IL) IL-6 and IL-1β), anti-inflammatory cytokines (IL-10), and chemokines (CCL2, CXCL1, CXCL10, and CXCL12), astrocytes can control microglia differentiation and macrophage activation [52,53,54]. These cytokines lead to hyperplasia of astrocytes, which results in the expression of inflammatory factors that can lead to reactive gliosis and scar formation. Astrocytes can directly restrict the entry of peripheral immune cells through the BBB. They are also the source of MMPs, a family of extracellular proteinases that degrade TJs and the ECM after ischemia, leading to the detachment of astrocytic end feet [55].

### 2.5. Microglia and Macrophages

Microglia are resident CNS macrophages that originate from the mesoderm during embryonic development and are widely distributed within the CNS. However, the basal ganglia and cerebellum have a higher abundance of microglia than the cerebral cortex [56]. They migrate into the brain and are termed “resting microglia” due to their low phagocytotic properties [57]. These cells communicate with endothelium to help regulate the BBB. As the primary immune cells that account for ~5–15% of all cells in the human brain, microglia can undergo morphological changes that allow them to phagocytose and produce pro-inflammatory cytokines IL-1 and IL-6, and enhance the expression of Intercellular Adhesion Molecule-1, P-selectin, and E-selectin [55,58]. As a result, the accumulation, migration, and adherence of leukocytes across endothelium allow them to mediate inflammatory cascades that further exaggerate the level of infarction.

During disease/trauma microglia become activated, and the extent of their activation is correlated to the severity and type of brain injury [59]. Activation of microglia is associated with dysfunction of the BBB via changes in TJ protein expression and increased BBB permeability [2]. High levels of neurotoxic mediators such as nitric oxide, peroxide, inflammatory cytokines (i.e., Tumor Necrosis Factor-α (TNF-α)), and proteases, as well as complement components [59,60], are produced, ultimately leading to cell injury in the CNS and neuronal cell death. Microglia are adept at sensing any small disturbance in the BBB [61,62] and maintaining BBB integrity during inflammation.

### 2.6. Junctional Complexes

ECs form the inner lining of blood vessels, creating a barrier between vessels and tissues. Lateral spaces between adjacent ECs called TJs, or Zona Occludins (ZO), and their proteins control the low paracellular permeability and high electrical resistance of the BBB [57]. Common transmembrane TJ proteins include Claudins (primarily Claudin-5) and occludin, which are phosphoproteins with four transmembrane domains that span the intracellular cleft, binding to proteins on adjacent ECs. Claudins and occludin are associated with cytoskeletal signaling proteins such as ZO-1 and ZO-2 and link TJs to the primary cytoskeleton proteins like actin for the maintenance of structural and functional integrity of the endothelium [63]. Junctional adhesion molecules are another family of transmembrane proteins that have a single transmembrane domain and are located at the borders of endothelial cells. These molecules are involved in cell-to-cell adhesion and leukocyte transmigration across the BBB [64]. The regulation of polar solutes and macromolecules across the barrier prevents the passage of unwanted and potentially damaging material such as peptides and proteins between the blood and brain. Through mechanoreceptor properties, ECs can respond and adjust vascular resistance via vasodilation and constriction to compensate for alterations of perfusion pressure and maintain a relatively constant CBF and microvascular pressure that contribute to cerebral autoregulation [65]. Akin to TJs, adherens junctions are protein complexes that occur at cell-cell junctions in endothelial cells. Unlike TJs, AJs join and maintain the connection between actin filaments of the cytoskeleton of neighboring cells. Transmembrane proteins called cadherins are a group of proteins that bind with other cadherins on adjacent ECs to help ECs stick together and regulate the intracellular signaling pathways that control gene transcription [66]. Additionally, gap junctions are intercellular channels that allow ions and molecules to pass through resulting in changes in membrane potential from one cell to another. Contrary to the extremely low permeability of tight junctions, gap junctions allow for the passage of certain molecules between cells. Consisting of connexin proteins, these structures allow for rapid propagation of action potentials along with the slow diffusion of nonorganic ions, secondary messengers, and other small water-soluble molecules [58]. They also transmit chemical signals and metabolites between cells, aiding in the function of vascular, neuronal and glial tissue [58]. Degradation of gap junctions contributes to the release of inflammatory mediators and disrupts the homeostasis of the CNS environment as exemplified under condition of ischemic stroke. These intercellular junctions are important for providing efficient and selective barriers against undesirable environmental conditions, providing the structural integrity of the cells that make up this barrier, and the overall homeostasis of the brain.

### 2.7. Basement Membrane

The basement membrane (BM) or ECM is an amorphous structure located on the abluminal side of endothelial cells or basal side of epithelial cells [67,68], and is comprised of multiple components such as laminins, collagen, nidogen, and heparan sulfate proteoglycans [67,68,69,70], forming a close contact with the endothelium. The BM plays a crucial role in maintaining vascular integrity and providing a rigid support to vessels and surrounding cells [70] by surrounding the capillaries and separating them from neighboring astrocytes and pericytes [71]. The ECM of the basement membrane limits the transmigration of red blood cells during hemorrhage and leukocytes during inflammation. Dysfunction and degradation of the BM are associated with several neurological disorders and increased barrier breakdown and edema [72,73]. Interestingly, endothelial cells, astrocytes, and pericytes are known to synthesize and deposit specific laminin isoforms in the BM that modulate BBB function [74,75].

## 3. Processes and Signaling Pathways Involved in the Development and Remodeling of NVU

During the development of the brain vasculature, blood vessels are formed from intraneural endothelial cells and undergo extensive expansion and branching to form nascent (leaky) blood vessels via vasculogenesis and angiogenesis. Then the perivascular cells differentiate and mature, remodeling into the BBB with highly selective permeability properties. Vasculogenesis refers to the proliferation and differentiation of endothelial cells from mesoderm-derived angioblasts to form the primary vascular plexus throughout the body [76], while angiogenesis refers to the more convoluted vascular complex that follows the formation of the primary vascular plexus via branching from the pre-existing vessels in which the vascular network of the brain is predominantly formed [77]. The formation steps are mediated by endothelial and neural progenitor cells, while the maturation steps are promoted by pericytes during embryonic development and later maintained by astrocytes throughout adulthood. These processes require precise coordination between cell signaling and behavior and are tightly regulated by signaling molecules and pathways. However, in the adulthood following steno-occlusion, vascular remodeling takes on new forms to accommodate the demand for increased flow acutely and chronically, which is discussed in detail in the following sections.

### 3.1. Formation of Blood Vessels (Vasculogenesis and Angiogenesis)

The formation of initial blood vessels is regulated by many angiogenic factors including Vascular Endothelial Growth Factor (VEGF), Wnt, G-Protein Coupled Receptor 124 (GPCR124), and Transforming Growth Factor-β (TGF-β).

Predominantly produced by neural progenitor cells, VEGF directs the migration and proliferation of ECs from angioblasts [78]. It is also essential for the entry of blood vessels into the developing neural tube and retinal vascularization in various vertebrate species [79]. Six homologs of VEGF have been identified, and each has a separate role in angiogenesis [76]. When bound to its main signal transducing receptor tyrosine kinase Flk1 (VEGF receptor 2) in endothelial progenitor cells, it is required for endothelial cell survival and the induction of brain angiogenesis [80]. In a study of mice deficient for the VEGF receptor 2, there was no observed blood vessel formation throughout the body and early embryonic lethality with severely compromised blood vessel formation in ligand deficient heterozygous VEGF+/− and homozygous VEGF−/− mice [81]. Under physiological conditions, VEGF is downregulated to maintain the balance between pro- and anti-angiogenesis, however, during diseased or traumatic states, VEGF plays a role in BBB breakdown and increased endothelial permeability along with vascular sprouting [76]. Although VEGF has been observed to induce the growth of new vessels, these vessels are immature and leak, which may exacerbate the edema caused by various brain injuries. Nonetheless, VEGF is essential in signaling initial blood vessel formation of the primary vascular plexus and the direction of sprouting.

In addition, Wnt also initiates brain angiogenesis and is important for BBB formation and maintenance. Mainly made by neural progenitor cells during early embryonic development, Wnt ligands bind with frizzled receptors on the endothelium, a step that is necessary for the migration of endothelial cells into the embryonic neural tissue and the establishment of TJs [3]. When bound to its receptors in endothelial cells, it inhibits β-catenin (an effecter molecule of the Wnt pathway) degradation and promotes β-catenin translation, which leads to the transcription of target genes [3,82]. B-catenin has a role in embryonic vascular development and is expressed only in developing CNS vasculature [83,84]. It also leads to the transcription of BBB-related genes such as those encoding for Glut-1 and TJ molecules [3]. It is the same signal that drives EC migration into the CNS that also induces the BBBs functions. Conditional knockout of β-catenin in ECs leads to a reduction in CNS vessels and vascular hemorrhage and malformation [76], highlighting the importance of the Wnt pathway.

Moreover, GPCR124 is an essential endothelial receptor for brain-specific angiogenesis. It is mainly expressed in endothelial cells and pericytes from embryogenesis to adulthood. It regulates the migration of endothelial cells and the expression of Glut-1 [3]. Several studies have revealed that GPCR124 is essential in endothelial cells for proper neural tube vascularization. GPCR124 knockout mice are embryonic lethal due to defects in the vasculature of the developing CNS [3,85]. The resulting impaired EC survival, growth, and migration results in the inability of angiogenic vascular sprouts to invade the embryonic neuroectoderm [86]. It should also be noted that GPCR124 knockout and Wnt signaling mutants have similar vascular defects such as the lack of Glut-1 expression [87], indicating that they may interact during development. On the other hand, VEGF is unaffected in the absence of GPCR124, suggesting little to no interaction between the two.

Furthermore, TGF-β is expressed in neural progenitor cells and glial cells during development. Once secreted by astrocytes and CNS ECs, it exists in a latent form. Its activation is mediated by the interaction between latency associated peptide (LAP) and integrin-avβ8, leading to the proteolytic degradation of LAP and the release of active TGF-β. This may in turn initiate GPCR124 expression for proper neurovascular development. While GPCR124 may function and be regulated independently from VEGF, TGF-β in endothelial cells, it may initiate the expression of GPCR124 [88]. TGF-β-signaling between endothelial cells and pericytes contributes to BBB induction by promoting pericyte–endothelium interaction and barrier formation in the BBB through their adhesion. Both ECs and pericytes express TGF-β and its receptor TGF-βR2 [89]. The signaling of TGF-β in pericytes leads to the production of ECM components such as proteins that contribute to BM formation and overall enhance BBB integrity [89]. TGF-β signaling in ECs drives pericyte adhesion through the upregulation of Cadherin-2 (N-cadherin), which leads to tight adhesion between ECs and pericytes [3].

### 3.2. Maturation of Blood Vessels (Barriergenesis)

Pericytes and VSMCs are mural cells located at the abluminal surface of vessels, respectively. Barriergenesis involves the recruitment of mural cells and the generation of the extracellular matrix, which forms the BBB. Newly formed blood cells in the brain are leaky, hinting to the fact that the barrier properties of the BBB arise later in development. Mural cells aid in the development of the BBB by supporting sprouting vessels, along with the differentiation and maturation of ECs. Astrocytes are recruited and integrate endothelial–astrocyte interactions needed in maintaining the BBB properties during cerebral development and adulthood.

Pericytes are recruited by endothelial cells of nascent blood vessels via Platelet-Derived Growth Factor (PDGF) signaling pathway PDGFβ/PDGFβR. Pericytes that express the receptor PDGFR-β are recruited to the endothelial surface resulting in pericyte proliferation and co-migration with sprouting vessels [3]. PDGFβ/PDGFβR actively regulates BBB induction and recruits pericytes to ECs and nascent cerebral blood vessels, in addition to aiding with vessel maturation via the recruitment of mural cells. PDGFRβ signaling through PDGFRβ is imperative for pericyte generation [90], and therefore decreased BBB permeability. Mice deficient for PDGFβR or its ligand PDGFβ lack brain pericytes, which ultimately leads to embryonic lethality before or at birth, CNS microhemorrhages, and morphological signs of increased vascular permeability [87]. Interestingly, the reduction of pericytes was only observed when receptor expression levels were reduced [90]. Additionally, ECs have an atypical distribution of junctional proteins and increased vascular permeability [3]. This demonstrates the inverse correlation between pericyte coverage and BBB permeability. Pericytes also modulate barrier properties and the integrity of the BBB using ECM proteins synthesized by astrocytes such as laminin, specifically laminin-α2β1γ1 (laminin-211), by depositing it into the basement membrane, which is integral for promoting brain vascular smooth muscle cell differentiation and maintenance [91]. These cells ultimately play an important role in the strength and integrity in blood vessels, and therefore the maintenance of TJs and BBB maintenance.

Astrocytes are glial cells that ensheathe blood vessels with their end feet. They carry out neurotransmitter clearance and recycling, provide the nutrition needed for neurons, and regulate the extracellular potassium balance and the BBB. Astrocytic signaling and interaction lead to the development of more advanced TJs and BBB permeability. This sealing of inter-endothelial TJs through the upregulation and redistribution of TJ proteins occurs during maturation and then needs to be maintained throughout life. The BBB is formed (at E15) before astrogliogenesis occurs (at E18) [82], so it is thought that astrocytes maintain, as opposed to induce BBB properties. Wnt signaling is involved in both brain angiogenesis and BBB maintenance. Astrocytes are a primary source of Wnt during postnatal development and adulthood [92]. They actively contribute to the maintenance of the barrier properties of BBB and BBB transporter expression.

In addition, Shh (Sonic hedgehog) signaling is required for barrier maintenance of the BBB and limiting BBB permeability by enhancing the expression of TJ proteins. Astrocytes are also the major transducers of Shh in the brain. Shh binds to its receptor Patched and induces a depression of Smoothened, an integral transmembrane protein involved in the hedgehog signaling pathway critical in embryonic development and tissue maintenance as an adult [93]. It should be noted that Shh in healthy adult mammalian brains is expressed by neurons [94], denoting Shh signaling as a mediator of neuron-astrocyte communication. It is also needed for vascular tube formation. In a study focusing on the effects of Shh on the formation of micro vessel-like structures, it was found that after 24 h from treatment, Shh lead to a significant increase in microvessel-like structures compared with untreated cultures [95]. In addition, when an inhibitor of hedgehog signaling was used in the treatment, the formation of micro vessel-like structures was not observed [95]. Shh exerts positive effects on the expression and secretion of factors such as VEGF and angiopoietins and other proangiogenic factors such as cytokines.

Likewise, Src-suppressed C-kinase substrate (SSeCKS) is expressed in astrocytes and enhances expression of Angiopoietin-1 that binds to Tie2 receptors on endothelial cells resulting in increased TJ protein expression, decreased vascular permeability, and the promotion of vessel stabilization via recruitment of smooth muscle cells and pericytes. In a study observing the defects observed in mice lacking Angiopoietin-1 (Ang-1), endothelial cells were found to be poorly associated with the underlying matrix elements which may be able to explain other defects such as failure in branching, vessel segmentation, and overall vessel integrity [96]. Angiopoietin is an angiogenic factor during embryonic vessel development and a maturation factor. In addition to decreasing the expression of VEGF through the reduction of Activating Protein-1, a transcription factor that stimulates the expression of Ang-1 acting as an anti-permeability factor in astrocytes [44]. Interestingly, in the absence of pericytes, the expressions of genes known to increase vascular permeability such as Angiopoietin-2 are upregulated. SSeCKS was also found to increase the expression on ZO-1, ZO-2, and claudin-1, supporting the theory that SSeCKS expression helps signal for BBB maturation. SSeCKS regulates BBB differentiation by adjusting both brain angiogenesis and the formation of tight junctions [95]. These findings support the hypothesis that SSeCKS expression acts as an activation signal for BBB maturation.

Conversely, Apolipoprotein E (ApoE) is a cholesterol and phospholipid transporter molecule produced by astrocytes and involved in the metabolism of fats that astrocytes use to regulate the BBB. Targeted replacement of murine ApoE with the human ApoE-3 isoform showed an intact BBB under multiphoton microscopy, while those with ApoE-4 had apparent BBB disruption [97], suggesting different effect of ApoE-3 and ApoE-4 on BBB permeability [3]. Specifically, ApoE-3 inhibits the inflammation pathway caused by ApoE-4 which results in greater barrier function of the BBB. ApoE-4 activates the CypA-NFKB-MMP9 inflammatory pathway in pericytes which leads to BM and TJ protein degradation [3,97]. In another study on the effects of ApoE-3 and ApoE-4, researchers were able to detect reductions in TJ and BM proteins in ApoE-4 mice at just 2 weeks of age [97]. They later found that these measurements progressively increased with age. The distinct differences between these isoforms are crucial in developing treatments without causing further damage.

### 3.3. Arteriogenesis

The collateral circulation is a unique blood vessel present in the microcirculation of most tissues connecting a small fraction of the outer branches of adjacent arterial trees [98,99]. Collateral circulation protects against ischemic brain injury through the alternate routes of perfusion [100,101]. Furthermore, endothelial and smooth muscle cells of collateral vessels have morphologic and functional differences from nearby similarly sized arterioles [102]. Arteriogenesis refers to an anatomic increase in lumen diameter and wall thickness of the collateral vessel induced by occlusion or stenosis in the artery [103,104]. It is usually accompanied by an increase in collateral length, resulting in increased tortuosity and requires days to weeks for the final new diameter and length to be achieved [105]. Arteriogenesis is induced by increased Fluid Shear Stress (FSS) [101,106] caused by the mechanical stress of blood flow over the surface of the vessel and acts tangential to the vascular wall. Following the gradual occlusion of the main feeding artery, an increased pressure gradient between the pre and post-stenotic perfusion territories shifts blood flow to the small collateral anastomoses, bypassing the stenosis and causing the FSS on the collateral endothelium to increase [107]. Increased FSS will lead to the activation of the endothelium, monocyte invasion, and the secretion of growth factors and cytokines, followed by matrix digestion [108]. These changes lead to the proliferation of the endothelial and smooth muscle cells, and ultimately the remodeling of collateral vessels to accommodate increased blood flow [109].

Arteriogenesis is predictive of improved long-term clinical outcomes in patients treated with and without thrombolysis for stroke [98,101,110]. Risk factors such as metabolic syndrome and diabetes are associated with impaired coronary collateral vessel formation via arteriogenesis [111,112]. Metabolic syndrome, hyperuricemia, and age were found to be associated with poor leptomeningeal collateral status in patients with acute ischemic stroke [113,114,115]. Failure of collateral blood flow is associated with infarct growth in patients with ischemic stroke [116]. An experimental study suggests that collateral failure is more severe in aged rats with significantly impaired pial collateral dynamics compared with younger rats [117]. A dramatic increase in intracranial pressure (ICP) was often observed in animals around 24 h after the onset of even a small-scale stroke [118], correlated with a significant linear reduction of collateral blood flow in response to progressive ICP elevation [119,120]. Thus, a similar transient ICP elevation occurring during the first 1 to 2 days post-stroke is a likely mechanism to explain the delayed infarct expansion in patients with minor stroke.

Taken together, vascular risk factors may exacerbate the outcome of stroke patients by contributing to poor collateral status, thus could be a target of intervention. Therapeutic stimulation and acceleration of this natural protective mechanism are also becoming a target to improve blood supply to the ischemic tissue. Several experimental studies have shown that the speed of arteriogenesis is not limited to its natural time course [121]. The infusion of transforming growth factor-beta1, fibroblast growth factors, or granulocyte-macrophage colony-stimulating factors (CSFs) into the peripheral or coronary collateral circulation led to a significant increase in collateral conductance compared with untreated animals [122,123,124]. In a rodent study, Niaspan, a drug increasing tumor necrosis factor-α-converting enzyme expression and CSFs significantly elevated local cerebral blood flow and increased arteriogenesis in the ischemic brain after stroke [125,126].

## 4. Remodeling of NVU after Ischemic Stroke

### 4.1. The Effect of Stroke on Endothelial Cells in the NVU

Increased vascular permeability is a hallmark of stroke accompanied by TJ disruption occurring in a stepwise fashion involving all members of NVU (Table 1). In an ischemia/reperfusion model of transient Middle Cerebral Artery Occlusion (tMCAO), BBB permeability exhibited a biphasic manner with an increase occurring at 3 and 72 h after reperfusion, and changes in claudin-5, occludin, and ZO-1 protein levels [127]. A later study confirmed that following a mild tMCAO, TJs were stable during the first 24 h after reperfusion, but then underwent significant breakdown and remodeling 48 to 58 h after reperfusion in claudin-5 reporter mice expressing eGFP [128], showing stepwise recruitment of transcellular and paracellular pathways underlying stroke-induced BBB breakdown. The study highlighted caveolin-1 (cav-1)-mediated transcytosis as the initiating event in the early phase after stroke leading to subsequent TJ remodeling in the later phase. Under physiological conditions, caveolae-mediated transcytosis across ECs is suppressed by the major facilitator superfamily domain containing 2a (MFSD2A), which is selectively expressed in brain ECs [129]. By acting as a lipid flippase, MFSD2A prevents the formation of caveolae vesicles [130]. During the early phase of reperfusion, ischemia-induced VEGF stimulates caveolae-mediated transcytosis by increasing cav-1 expression, leading to non-selective vesicular transport of blood-borne molecules across ECs. In the later phase, the breakdown of the vessel wall and TJ assembly further exacerbate BBB dysfunction [128,131,132].

Additional stressors such as oxidative stress cause further vascular impairment including vascular reactivity by affecting the bioavailability of Nitric Oxide (NO) [149,150,151]. Oxidative stress also induces cav-1 phosphorylation, and thus caveolae-mediated transcytosis [152]. Activation of small GTPase RhoA, and Rho-Associated Coiled-coil Kinase (ROCK) in brain ECs via stroke promotes the association between endothelial NO Synthase (eNOS) and cav-1 [134], leading to eNOS inhibition following its translocation to the caveolae compartments [153,154,155]. ROCK also directly inhibits the expression of eNOS by reducing its mRNA stability [156]. Thus, pharmacological blockade of ROCK reduces oxygen-glucose deprivation-induced hyperpermeability [157], and knocking out ROCK2 enhances eNOS expression and reduction of infarct size [158].

The consequences of ischemia-increased permeability include leakage of toxic compounds into the brain and dysfunction of specific transporter systems for ions and other molecules such as glucose [159,160] (Figure 2). This chronic BBB leakage [161] is also observed in other neurodegenerative disorders. Notably, changes in brain ECs following BBB disruption appear to be conserved across CNS diseases and injuries. A recent large-scale study investigating EC transcriptomic changes revealed a core BBB dysfunction module shared by four different brain injury models, namely permanent MCAO, TBI, experimental encephalomyelitis, and kainite-induced seizure [162]. They found that these commonly induced genes are involved in the inflammatory response, leukocyte migration, adhesion, and angiogenesis. Interestingly, brain ECs in each disease model acquired a peripheral endothelial gene expression profile with a leakier phenotype.

### 4.2. The Effect of Stroke on Pericytes in the NVU

With their ability to regulate capillary vessel diameter via Ca^2+^ signaling and inhibit the expression of genes promoting vessel permeability, pericytes play an essential role in modulating blood flow after stroke, as well as maintaining vascular stability via the secretion of BM proteins [163]. During the first 2 h after occlusion of the internal carotid artery in cats, pericytes were found to migrate away from brain microvessels [135], eventually leading to a loss of pericyte coverage due to their detachment from the microvessel walls and cell death. In a later study using photothrombotic stroke in mice, it was postulated that the rapid activation of MMP-9 secreted from pericyte somata degraded underlying TJ complexes resulting in plasma leakage between the pericyte somata and the bordering capillary wall [138]. Taken together with the processes involved in endothelial cells’ response to stroke, pericyte activation and secretion of MMP-9 represents an intermediate step between leakage by transcytosis (a form of transcellular leakage) and eventual TJ degradation (paracellular leakage). This implies that pericytes may use MMP-9 to actively migrate from the endothelium to the impacted area to participate in post-stroke revascularization and repair of the NVU [138]. Meanwhile, stroke-activated PDGFR-β and Ang1/Tie2 signaling pathways may enhance pericyte survival and in turn, induce the expression of TJ proteins in ECs to remodel the NVU [164]. Another result of the cross-talk between pericytes and ECs is the induction of MFSD2A in ECs by pericytes, inhibiting caveolae-mediated transcytosis and BBB leakage [11,35,129,165,166,167].

In addition to PDGF and Angiopoietin signaling, pericytes also respond to neuron-activating signals including NO, glutamate, and Prostaglandin E2 (PGE2) [29,168,169]. Transient stimulation by hypoxia induces pericyte relaxation and dilation of capillary vessels [31,136], while sustained hypoxic-ischemic injury results in oxidative stress and pericyte constriction, and eventual death [29,137]. Loss of pericytes not only impairs BBB function and blood flow regulation but also affects neuronal survival. A recent study showed that pericyte loss led to circulatory failure and neuronal loss by depletion of Pleiotrophin (PTN), a pericyte-secreted growth factor [139].

### 4.3. The Effect of Stroke on Astrocytes in the NVU

Due to the direct contact of astrocytic end feet with neurons and capillary vessels, astrocytes not only regulate cerebral microvasculature and functional flow but also neuronal survival and repair, via gap junction proteins that connect astrocyte networks into a functional syncytium [13,49,50,51,170]. Astrocytes are heterogeneous and regionally specified [171,172,173]. By expressing gap junction proteins, neurotransmitter receptors, transporters, and ion channels, astrocytes are versatile in function. Following permanent MCAO, astrocytes are activated by trophic factors produced by pericytes located within the ischemic regions, leading to peri-infarct astrogliosis [145], with upregulated Glial Fibrillary Acidic Protein (GFAP), an intermediate filament serving as a biomarker for astrocyte activation [141]. Astrocytes are also known to interact with microglia after stroke.

In the peri-infarct cortex after stroke, propagating waves of neuronal or astrocytic origin called Cortical Spreading Depolarization (CSD) can further decrease blood flow and worsen stroke outcome [174,175,176]. Specifically, CSDs can induce vasoconstriction of vascular smooth muscle cells by increasing astrocytic vasoconstrictor 20-hydroxyeicosatetraenoic acid [143]. Astrocytic gap junctions have also been implicated in propagating CSDs and leading to exacerbated brain damage [144]. Following ischemia, astrocytes are also known to secrete pro-angiogenic factors that promote the growth of new capillaries toward the infarcted tissue [140]. By secreting neurotrophic factors, astrocytes can minimize ischemic damage to neighboring cells through the formation of glial scars [137,142]. Paradoxically, glial scars appears to be detrimental to functional recovery by acting as a barrier to neuronal regeneration [177].

### 4.4. The Effect of Stroke on Microglia and Macrophages in the NVU

Emerging evidence suggests that Purinergic receptor P2Y, G-protein coupled, 12 (P2RY12), a resting state microglia marker, is required for the rapid closure of the BBB [148]. Although some studies suggested that microglial ablation in the mature mouse brain did not affect BBB function [178,179], other studies showed that mice treated with the P2RY12 inhibitor clopidogrel, as well as those in which P2RY12 was genetically ablated, exhibited significantly diminished movement of juxtavascular microglial processes and failed to close BBB openings caused by the laser. Microglia dynamically transition into a reactive state after stroke [180,181], owing to the initial leakage of blood serum components such as fibrinogen induces local activation of microglia. Finely tuned to detect small disturbance in the BBB [61,62], the recruitment of microglia to blood vessels occurs within 6 h of reperfusion, with a significant accumulation in perilesional tissue. After 24 h of reperfusion, microglia fully enfold the small blood vessels in the peri-infarct region [147]. Individual perivascular microglia display intracellular vesicles containing CD31-positive inclusions. This suggests the incidence of phagocytosis of brain ECs, is correlated with BBB breakdown as demonstrated by the extravasation of Evans blue from perfused vessels. At 72 h post-MCAO, there was complete blood vessel degradation, and the remaining vascular debris was cleared by microglia and invading immune cells [147]. Reactive microglia also secrete MMP-9 and MMP-3, proteases that can break down the basement membrane and contribute to further BBB leakage [146].

Studies have also established a crucial role of Perivascular Macrophages (PVMs) in the regulation of stroke inflammatory response, outcome, CBF, and vascular function. CD163 positive PVM infiltrates were found in ischemic brain lesions of post-mortem samples [182], suggesting their activation in response to ischemia. Sixteen hours after 60 min of transient MCAO, CD163-positive cells exhibited an upregulation of the HIF-1 pathway including VEGF, and an increase in genes encoding the ECM and leukocyte chemoattractants; while depletion of PVMs reduced granulocyte infiltration, BBB permeability, and VEGF expression [183]. Like other cells in the myeloid lineage, PVMs are phagocytic [184]. PVMs are also known to promote ROS-induced BBB degradation in a mouse model of hypertension. Depleting PVMs in this case, not only effectively reduced oxidative stress, but also restored neurovascular coupling, improved CBF, and ultimately reversed cognitive impairment [185].

### 4.5. The Effect of Stroke on the Basement Membrane

Following ischemic stroke, activated ECs and pericytes generate MMPs that degrade the BM [186,187]. Other proteinases including plasminogen activators, heparinases, and cathepsins are also dramatically enhanced and contribute to the degradation of the BM.. The proteolytic breakdown of ECM proteins such as laminin-5 or type IV collagen increases the permeability of BBB and exposes cryptic epitopes that promote EC and pericyte migration [188].

BM degradation was observed about 10 min after reperfusion, and the loss of BM was detected as early as 1–3 h after MCAO [189]. Morphologically, the well-defined electron-dense BMs become diffused and faint after ischemia [132,190]. After MCAO, both MMP-2 and MMP-9 are significantly upregulated resulting in the digestion of ECM proteins and BBB breakdown [191]. Cathepsin B and L proteases are also augmented following stroke and will degrade Heparan Sulfate Proteoglycans (HSPGs) [192]; HSPG Agrin is stabilizes adherens junctions in mouse brain ECs [193]. Therefore, reduction of HSPG will dramatically affect the BBB and increase permeability. As mentioned so far, proteases activated during ischemic stroke have detrimental effects during the acute phase. However, in contrast, they play an important role in angiogenesis and vascular remodeling in the late phase [187].

## 5. Predisposition of NVU to Vascular Risk Factors

Most stroke patients have associated risk factor(s), which have a significant impact on stroke outcomes. We may be able to reduce stroke occurrence and improve the efficacy of stroke therapies by understanding the implications of vascular risk factors to the NVU (Table 2). Therefore, in this section, we review the predisposition of the NVU to vascular risk factors, including diabetes, hypertension, and hyperlipidemia, and non-modifiable risk factors, such as age and sex.

### 5.1. Diabetes and Hyperglycemia

In diabetic female mice, astrocytic foot processes were detached and microglial cells were indicated to play an invasive damaging role in the injury [208]. This detachment may result in impaired functional hyperemia, decreased energy substrate supply, and neuronal dysfunction [208]. Pericyte apoptosis has also been recognized in vivo in hyperglycemic rats and in vitro in retinal cultures [209]. Furthermore, endothelial cell deterioration at the electronic-microscopic level was observed in diabetic mice: loss of electron density, basement membrane thickening and rearrangement, increased transcytotic-pinocytotic vesicles, and aberrant mitochondria [208].

Enhanced signal intensity on MRIs further suggests an increased BBB permeability in diabetic patients [194,195], consistent with BBB dysfunction observed in animal and cell culture models of diabetes [197,198,199,203]. TJ disruption is a significant anatomical change in diabetes accounting for impaired BBB integrity. Protein levels of TJ components, such as occludin, claudin-5, and ZO-1, decrease after hyperglycemia [202,205,206]. Therapeutic agents that prevent TJ protein alterations protect BBB integrity in diabetic animals [207]. Hyperglycemia induces the activation of upstream signaling molecules and subsequent development of oxidative stress that play a major role in TJ loss [196,200]. Reactive Oxygen Species (ROS) and increased MMP activity are also suggested to mediate BBB breakdown after hyperglycemia. This is supported by evidence that ROS and MMP inhibitors preserve TJs and improve BBB integrity [201,204].

### 5.2. Hypertension

Research suggests the combined effects of increases in shear stress, endothelial dysfunction [227,228], Angiotensin II (ANG II) [210,229], and inflammatory signals affect vascular dysfunction in hypertension. Chronic increases in intraluminal pressure cause vessel wall thickening and biomechanical changes to the NVU [230,231]. Furthermore, ANG II has been shown to cause deleterious changes in multiple cell types of the NVU. Chronic ANG II infusion increased Transient Receptor Potential Vanilloid-type 4 (TRPV4) channel expression in astrocytes and enhanced TRPV4-mediated currents and astrocyte Ca^2+^ events [216]. Moreover, chronic ANG II infusion increased astrogliosis and hippocampus microglial activation. These studies indicate that the cells of the NVU are subject to alterations in intracellular Ca^2+^ during hypertension and increase their vulnerability.

BBB abnormalities occur in both acute and chronic hypertension although they began at the early stages of hypertension [212] In laboratory animals, BBB impairment was observed in the cerebral cortex and deep gray matter at five months and the hippocampus as young as three months of the Spontaneously Hypertensive Rats (SHRs) [211,214,232,233]. Acute hypertension due to aortic constriction above the renal arteries increased BBB permeability 8 days after surgery, coincided with an increased oxidative stress [213]. Mechanistically the alterations in EC junctions play a major role in the vascular anatomical changes underlying hypertension-induced BBB dysfunction. Chronic hypertensive rats demonstrate progressive morphological changes in BBB TJs, with increased loss of occludin and ZO-1 observed as early as four weeks [214], while acute hypertension leads to the loss of claudins [213]. Hypertension-induced inflammation and oxidative stress are known to exacerbate BBB permeability. In SHRs, pro-inflammatory cytokines such as IL-1β, IL-6, and TNF-α are significantly elevated at 8 months and oxidative stress starts to increase at 16 weeks [215,234].

### 5.3. Hyperlipidemia

Apolipoprotein E (ApoE) is essential for lipid and cholesterol metabolism and its absence results in hyperlipidemia [235]; thus, ApoE- deficient mice are the most used animal model for studying hyperlipidemia. At the age of 6–8 weeks, ApoE-deficient mice already exhibit significant BBB permeability [217]. One likely reason is that ApoE directly modulates BBB integrity by inducing TJ formation or suppressing inflammation in the NVU [97,218]. Another valid animal model is by injecting lipoprotein directly, which was shown in rats. Following the injection of human triglyceride-rich lipoprotein, leading to the releases of lipolysis products upon hydrolysis and increases in BBB permeability within 20 min [219]. Furthermore, some reports indicate the role of oxidative stress in increasing BBB permeability in hyperlipidemia [236]. Although several studies showed the relationship between hyperlipidemia and NVU dysfunction, the exact role of hyperlipidemia in BBB dysfunction remains largely unclear and warrants further studies.

### 5.4. Aging

Characteristics of an aged brain include reduced capillary density and cerebral blood flow, along with ultrastructural abnormalities in microvessels, such as microvascular fibrosis, basement membrane thickening, and loss of TJ proteins [237]. Regarding the latter, mice that are 24 months old have significantly less expression of occludin and ZO-1 than young adult mice [221]. In addition, the degeneration of pericytes in aging brains may also contribute to the compromised BBB integrity [222]. Increased BBB permeability in normal aging humans was reported as well [238]. For example, a study using advanced MRI to quantify regional BBB integrity further reveals that BBB dysfunction is a relatively early event in the process of an aging brain, originating in the hippocampus [222]. Early studies well documented age-related BBB changes: altered transport functions [220,239], increased glycosylation of microvessel proteins [240], and free radical damage [241]. Lastly, inflammation remains as an important mechanism in the age-related BBB permeability. Aged brains are in a low-grade but progressively inflammatory state, in which the state of immune cells is skewed towards a pro-inflammatory state [242]. This is supported by evidence showing increased amount of inflammatory mediators, such as IL-1β, IFNγ, and TNF-α, in the aging brain, with concomitant activation of microglia [221,243].

### 5.5. Sex

Reproductive hormones are the driver for the difference in BBB integrity between males and females in response to injury or diseases. For example, the decline in estrogen exhibited in aging female mice were found to be associated with increased BBB permeability compared with younger female mice [223]. The protection of BBB integrity observed in normal young females is likely an effect of estrogen, since ovariectomy in 3-month-old female mice induces the extravasation of Evans Blue into the brain [226]. Also, compromises in BBB integrity are observed in young adult male mice but not in young females [224]. It is further evidenced by the protection of BBB breakdown in old, reproductively senescent females or ovariectomy-operated young females by estradiol replacement [226]. Mechanistically, TJ proteins and their regulators are believed to be significant sites where estrogen affects BBB permeability. Estradiol treatment was observed to increase transendothelial electrical resistance in cultured brain ECs and upregulates claudin-5 [225]. Conversely, annexin, a central modulator of TJ integrity, is diminished in aged females but dramatically upregulated by estradiol [226].

## 6. Remodeling of NVU after Cerebral Hemorrhage

Hemorrhagic stroke consists of three main subtypes, i.e., Intracerebral Hemorrhage (ICH), Intraventricular Hemorrhage (IVH), and Subarachnoid Hemorrhage (SAH), which represent about 20% of stroke cases [170,244,245]. The average mortality rate of hemorrhagic stroke is approximately 50% and only about 20% of survivors regain functional independence at 6 months, whereas more than one-third of affected patients will not survive the first year [246,247].

Hemorrhagic stroke occurs when a weakened vessel ruptures, allowing blood to traverse the broken BBB into the brain tissue, or fissures, leading to a mass effect (primary damage) that increases the intracranial pressure and decreases the cerebral blood flow [248,249]. Therefore, the primary damage of hemorrhagic stroke in brain tissue is from occupation (hematoma), ischemia (tissue surrounding the hematoma), and toxin (degraded blood). The secondary damage develops in the late acute and subacute phases resulting from the local and systemic responses to hematoma formation and its toxic effect on adjacent brain tissue. The causes of secondary damage include (1) cytotoxicity of the blood; (2) excitotoxicity, due to the release of excitatory amino acids like glutamate from injured neurons; (3) spreading depression, a slow and short-lived depolarization wave that propagates through the brain; (4) hypermetabolism; and (5) oxidative stress and inflammation [250].

Two major signaling pathways are involved in post-ICH NVU remodeling, namely angiogenesis [151,251,252] and inflammation. The mRNA of the main angiogenic factor VEGF and its receptors were detected as early as 2 days after ICH. These levels peaked at 21 days and persisted for at least 28 days post-ICH [253]. Similarly, rats exhibited angiogenesis following a SAH stroke paradigm, and the expression of VEGF was induced by hypoxia resulting from vasospasm [251]. New vessels were also detected around the hematoma after ICH [253]. As such, several studies suggest the modulation of angiogenesis via altered expressions of VEGF and its receptors to be a potential strategy for promoting ICH repair [254,255]. The prime example is the treatment with Ginkgo biloba extract EGb761 enhanced VEGF expression, increased microvessel density, and promoted neuroprotection in mice subjected to ICH induced by collagenase injection [254]. Conversely, inhibition of VEGF expression negatively affected stroke outcomes [254]. For instance, inhibition of High-mobility Group Box 1 protein (HMGB1), a member of the Damage-associated Molecular-pattern (DAMP) family of proteins, resulted in reduced levels of VEGF and Nerve Growth Factor (NGF), which is associated with a reduction of neurological function recovery following ICH [255]. Beneficial effects of VEGF on brain edema following ICH were also reported [252]. Although brain edema following ICH is more serious than cerebral ischemia and often leads to poor prognosis, Chu et al. showed in their animal study that VEGF did not affect BBB permeability after ICH.

Blood components can induce an inflammatory response in the perihematomal area, including perihematomal leukocyte infiltration, microglia activation, elevations in proinflammatory cytokines and chemokines, and increased production and activation of MMPs [250,256,257,258,259,260,261,262,263], leading to increased BBB permeability. In addition, leukocytes can also modulate BBB function upon adhesion and transmigration across the endothelium [264]. An inflammatory response in the surrounding brain area occurs soon after ICH and peaks several days later in humans and animals [265,266]. Neutrophil infiltration develops within 2 days in rats, and activated microglial cells persist for a month [267].

### 6.1. Structural NVU Remodeling following Cerebral Hemorrhage

#### 6.1.1. Endothelial Cells/Tight Junctions (TJs) Disruption/BBB

Analogous to ischemic stroke, TJs are disrupted after a hemorrhagic stroke. TJ disruption is associated with increased vascular permeability and homeostatic changes in the neuronal microenvironment. BBB impairment may be seen as early as 10 min following SAH, can be maximal at 24 h, and persist for up to 7 days after [268,269,270,271]. Abluminal platelets may also be seen as early as 10 min after SAH with intraparenchymal platelet aggregates seen at 24 h [272]. The primary insult following BBB dysfunction may disrupt TJs, transporters, transcytosis, and leukocyte adhesion molecule expression of endothelial cells, which may lead to brain edema, ionic homeostasis disruption, and immune cell infiltration, consequently causing neuronal cell death [273].

After vascular disruption, blood enters the brain parenchymal initiating the process of cell excitotoxicity, cell edema, oxidative stress, and neuroinflammation, which further disrupts the BBB. Blood components (e.g., thrombin, hemoglobin, iron) and the inflammatory response to them play a large role in ICH-induced BBB dysfunction [274]. The first phase of BBB disruption may result in early phases of cell death due to physical injury, brain edema, and increased intracranial pressure. The blood in the brain parenchymal produces neurotoxic factors inducing thrombin, fibrin, and erythrocyte components [16,274]. Thrombin is the cascade product of prothrombin during hemostasis after a hemorrhagic stroke. The binding of thrombin to protease-activated receptor 1 induces secondary BBB disruption through phosphorylating Src kinases and activating microglia [275].

#### 6.1.2. Pericytes

Pericytes may undergo several changes due to hemorrhagic stroke. In the acute phase post-stroke, bleeding in the environment causes the death of pericytes resulting in BBB damage, pericyte-mediated inflammatory cascades, white matter impairment, and ultimately neural injury. During the recovery period, in situ pericytes are activated and differentiate into neurons, glia, and endothelial cells to repair the neural vascular network. Many pericytes are recruited to the lesion and contribute to BBB remodeling, thus facilitating NVU recovery [276]. The loss of pericytes also has a profound effect on neurotrophic-dependent neuronal survival. Genetic ablation of pericytes results in loss of PTN expression, a pericyte-secreted growth factor whose loss contributes to neuronal death [139].

#### 6.1.3. Astrocytes

In hemorrhagic stroke, astrocytes undergo important structural and functional modifications that can either be neuroprotective or detrimental to neuronal function [277,278]. ICH increases the length of primary astrocytic GFAP-positive processes in perilesional tissue, correlated with functional forelimb recovery [279,280]. Astrocyte survival has been correlated with neuronal survival [281]. Through their secretion of neurotrophic factors, along with perivascular stromal cells, astrocytes minimize damage to neighboring cells through the formation of glial scars [137,142]. However, glial scars can also be detrimental to functional recovery by acting as a barrier to neuronal regeneration [177].

#### 6.1.4. Microglia

Following a hemorrhagic stroke, microglia are dynamically activated [180,181]. Following ICH, the initial leakage of blood serum components such as fibrinogen induces local activation of microglia, leading to a rapid inflammatory response. Activated microglia develop into an M1-like phenotype resulting in the production of pro-inflammatory cytokines such as IL-1b, IL-6, IL-12, IL-23, TNF-α, chemokines, redox molecules (NADPH oxidase, phagocyte oxidase, inducible NO synthase), costimulatory proteins (CD40), and major histocompatibility complex II (MHC-II) [282]. In both clinical and experimental rodent models, during the acute phase of ICH, proinflammatory factors are present in the brain starting 3 h after ICH and peaking at 3 days. Due to their pro-inflammatory phenotype, M1 microglia are linked to short-term brain damage. Within 1 week, a microglial phenotypic switch occurs from M1 to M2 [54]. Conversely, M2-like microglia are associated with anti-inflammatory and phagocytic functions and assist in the clearance of the hematoma. Microglia can also be activated to the M2 polarization state by IL4/IL3, which produces anti-inflammatory mediators IL-10, TGF-β, and glucocorticoids [282]. Several factors have been implicated in inducing microglial polarization including, nuclear factor-kB (NF-kB), signal transducer and activator of transcription (STAT1–STAT6), HMGB1, as well as PGE2 [54].

#### 6.1.5. Neurons

Neuronal damage following ICH results from primary and secondary injury. The primary injury is mainly caused by the effect of mass that increases intracranial pressure and decreases cerebral blood flow. Secondary injury is mostly caused by hemoglobin and its oxidized product hemin from lysed red blood cells [283]. Hemoglobin and hemin increase brain water content, inflammatory responses, and neuronal cell death [284,285]. Both apoptotic and necrotic cells have been identified in animal models of ICH [286,287] and the perihematomal region after surgical evacuation in humans [288,289]. Non-apoptotic cell deaths such as necroptosis and ferroptosis have also been detected in ICH brains [290,291]. Zille et al. demonstrated that ferroptosis and necroptotic signaling induced by lysed blood cells is sufficient to reach a threshold of death that leads to neuronal necrosis, and that inhibition of either of these pathways can bring cells below that threshold of survival [283].

#### 6.1.6. ECM

Disruption of astrocyte-derived laminin expression in animal studies causes spontaneous hemorrhagic stroke in deep brain regions (basal ganglia). A similar phenomenon has also been shown in human patients. In the acute stage of ICH, the synthesis and secretion of MMPs increase in different cell types, including leukocytes, activated microglia, neurons, and endothelial cells [292]. The levels of MMPs peak within the first days in the brain and peripheral blood and remain high across the first week after hematoma development. The enhanced activities of MMPs also play a key role in the pathophysiology of secondary brain injury following ICH. The activation of MMPs (mainly MMP-3 and MMP-9) is responsible for the degradation of the neurovascular matrix through the digestion of the principal components of the basal lamina (collagen type IV, laminin, and fibronectin) surrounding blood vessels [293] and TJ proteins [294,295], leading to the damage of vascular walls, loss of vascular integrity, increased permeability, the development of edema, and the extravasation of leukocytes into the brain parenchyma [296,297]. MMPs also play a key role in mediating neural network remodeling and recovery after ICH by participating in neural stem cell migration, releasing, and activating pro-angiogenic and neurotrophic factors, vessel remodeling, myelin formation, and axonal growth [298].

The activity of MMPs can be inhibited in the plasma by α2-macroglobulin, and inside tissues by tissue inhibitors of metalloproteinases, which comprise a family of four protease inhibitors [299,300]. The inhibition of MMPs in the acute stage of ICH represents an attractive target to mitigate secondary brain injury, thereby improving clinical outcomes. However, modulation of MMPs is complex due to their pleiotropic and biphasic nature with multiple roles that grossly depend on the stage of the hematoma development. Experimental studies suggest that early and short-term inhibition of MMPs after ICH can be an effective strategy to reduce cerebral damage and improve the outcome, whereas long-term treatment may exacerbate the effects of ICHs.

### 6.2. Risk Factors for Hemorrhagic Stroke

The primary and secondary causes of hemorrhagic stroke include Cerebral Venous Thrombosis, rupture of vascular malformations, or Dural Arteriovenous Fistulae (DAVF), CNS vasculopathy/vasculitis, or hemorrhagic transformation of ischemic stroke [245]. ICH is primarily caused by uncontrolled hypertension [301,302] while SAH is mainly caused by vascular abnormalities, such as aneurysms, arteriovenous malformation, and neoplasm, etc. [301].

#### 6.2.1. Hypertension

Hypertension is the most important modifiable risk factor for stroke. There is a strong, direct, linear, and continuous relationship between blood pressure and stroke risk [302,303]. The risk of hemorrhagic stroke increases when hypertension is left untreated [245]. A high proportion of hemorrhagic stroke relative to ischemic stroke can be found in developing countries, where the burden of hypertensive disorders is greater than in more developed countries [304]. Even among those who are not defined as hypertensive, the higher the blood pressure, the higher the risk of stroke [305].

Hypertension promotes stroke through increased vessel wall shear stress, endothelial cell dysfunction, and large artery stiffness in the brain. Dysfunctional angiogenesis may occur in hypertensive elderly patients in the recovering penumbra following stroke [170,228,304,306]. Typical neuropathologic changes in patients with hypertension include the replacement of VSMCs in the tunica media with fibro-hyaline material, thickening of the vessel wall, and false microaneurysm formation (Charcot-Bouchard aneurysms), resulting in an increased susceptibility to vessel wall rupture [307,308]. The most affected vessels are those with diameters within the 50–700 μm range. These ruptures often occur near the origin of perforating vessels from the basilar, anterior, middle, and posterior cerebral arteries [308].

#### 6.2.2. Vascular Malformation

Vascular malformation is one of the most common causes of hemorrhagic stroke in people under the age of 40 [309]. Among vascular malformations, Arteriovenous Malformations (AVMs) and Cerebral Cavernous Malformations (CCMs) are the most common causes of ICH in young patients [310], while Dural Arteriovenous Fistula (DAVF) is a rare type of AVM and tends to occur later in life. 

##### AVM

AVMs are tangles of abnormal vessels shunting blood directly from arteries into veins [311], forming a vascular mass called a nidus that is deprived of true capillary beds [312]. Abnormally high blood flow through arterial-venous shunting has been suggested to contribute to brain AVM rupture [313]. A recent study using single cell RNAseq revealed the endothelial cell molecular signatures within each of the arteriovenous segments of AVM patients, suggesting that AVM vasculature contained expanded perivascular cell diversity with abnormal vascular patterning and increased inflammation. They concluded that the interplay between vascular and immune cells also contributed to brain hemorrhage [314]. Most brain AVMs (bAVMs) are sporadic, and somatic mutations in RAS/MAPK (Mitogen-Activated Protein Kinase) pathway genes are associated with intracranial and extracranial AVMs [315,316,317]. In contrast, only about 5% AVMs are familial. One common form of familial AVM is Hereditary Hemorrhagic Telangiectasia, caused by autosomal dominant mutations in TGF-β/BMP-9 signaling pathway genes such as *ENG*, *ACVRL1/ALK1*, and *SMAD4.* Capillary malformation–arteriovenous malformation, another familial form of AVM is caused by mutations in EPHB4-RAS-ERK signaling pathway genes: *RASA1* and *EPHB4.* One recent study systematically investigated the contribution of germline variants to bAVM and explore the critical molecular pathways underlying the pathogenesis of bAVM. The authors identified 4 variants in de novo bAVMs through enrichment analysis of genes with likely gene-disrupting (LGD) variants. Notably, ATP6V1B2 has recently been linked to severe epileptic encephalopathy and deafness, onychodystrophy, osteodystrophy, mental retardation, and seizures syndrome, suggesting the malformation in bAVMs as an extension of ATP6V1B2-related disorders. In addition, it was found that in de novo LGD or missense variants, the angiopoietin-like protein 8 regulatory pathways were enriched compared with the control. Specifically, a single nucleotide polymorphism in ANGPTL4 was found to be significantly associated with bAVM, highlighting the important role of angiopoietin-like protein pathways in bAVM pathogenesis. In gene-based rare variants, SLC19A3, whose expression in the brain is entirely related to blood vessels, was identified as a disease-associated gene that has also been associated with epileptic encephalopathy [318]. These findings suggest the underlying role of genes associated with CNS disease and developmental defects in cerebral vascular malformations. However, the interactions between somatic and germline variants require further investigation.

Another study found that many germline variants in genes were related to TGF-B/Notch signaling using similar strategies via whole exome sequencing of sporadic bAVMs to cluster and pinpoint pathways affected by germline mutated genes. Although the mutated genes differed among patients, the pathways they resulted in were the same. Pathways related to microtubule formation, cell adhesion, and vascular remodeling, and ion conduction in endothelial cells were affected [319], eluding that these pathways are involved in bAVM pathogenesis. Additionally, after selecting loci affected by rare variants and then those most likely related to rare disease onset, none were detected in any of the control exomes from healthy patients [319]. This led them to the hypothesis that bAVMs are a result of the co-existence of low-penetrance loci controlling different processes in cell maturation and differentiation. This study highlights the genetic heterogeneity of bAVMs along with the potential for diagnosis based on specific genes.

The risk of ICH in the untreated course of brain AVM patients is estimated at 2–4% per year overall and 1–2% per year for unruptured AVMs [320,321,322]. The biggest predictors of subsequent hemorrhage from AVMs are initial hemorrhagic presentation, deep brain location, increasing age, exclusively deep venous drainage [323], and the presence of associated aneurysms [324]. Analysis of resected brain AVM tissue indicates that angiogenic and inflammatory pathways are involved in brain AVM pathogenesis [325,326]. Recent bodies of work have shown elevations in VEGF or alterations in the vascular wall and loss of pericytes contribute to bAVM rupture [327,328,329]. The abnormal vessels in human bAVMs vary in their degree of exposure to increased intraluminal flow and Venous Hypertension (VH). A positive correlation between VH and angiogenic activity was first proposed by Lawton et al. [330]. It has also been shown that VH upregulates the expression of VEGF and Hypoxia-inducible Factor 1-alpha (HIF-1α) [331,332]. Animal models suggest that an increased level of VEGF may contribute to the hemorrhagic tendency [327,333]. The creation of VH in mice with brain AVMs caused severe hemorrhage in the brain AVM lesions and high mortality [327]. Abnormal mural cell coverage is another characteristic that is linked to brain AVM hemorrhage. Both human and mouse brain AVM vessels have less mural cell coverage compared with normal brain vessels [328,329,334]. The number of pericytes is inversely correlated with the degree of hemorrhaging or clinically occult microhemorrhaging [328,334].

##### CCM

CCMs are a specific subset of vascular malformation characterized by clusters of dilated, thin-walled vascular channels lined by simple endothelium. They are low flow, angiographically occult lesions often associated with neurological morbidity and hemorrhage. Ultrastructural studies revealed abnormal or absent blood-brain barrier components, poorly formed tight junctions resulting in gaps between endothelial cells, lack of astrocytic foot processes, and a limited number of pericytes [335]. A study of sporadic human CCM lesions identified differential editing events pertaining to CCM-ECs compared with human brain microvasculature endothelial cells (HBMECs) via whole RNA sequencing data. They found that out of the 60% of genes that were edited in both HBMECs and CCM-ECs, greater than 37% of them were totally lost in CCM-ECs while 30% of them were partially lost [336]. Many of these differentially edited genes encode for proteins that are involved in pathways related to cell adhesion, angiogenesis, apoptosis, cell survival and cytoskeleton regulation [336,337]. Notably, large clusters of genes involved in inflammatory pathways were enriched in CCM-ECs. A similar study using transcriptome analysis on ECs isolated from CCM specimens identified a few dysregulated pathways involved in CCM onset. Both studies allude to the importance of the non-canonical Wnt5a pathway in CCM disease. One molecular cascade called the Wnt/Ca^2+^ pathway is involved in vasculature remodeling. The other pathway is the Wnt/planar cell polarity pathway and is involved in cell polarization along with TJ maintenance, and cytoskeleton organization. Primary cilia disassembly following fluid shear stress stimuli was recently proposed as a contributor in CCM signaling activation. Both studies reiterate that the impairment of CCM genes are not the only condition leading to CCM pathogenesis, however these specific genes that could potentially be the targets of new selective therapies.

CCMs can occur in a sporadic or familial form; familial cases often present with multiple lesions that grow in number and size over time, reflecting the dynamic nature of these lesions [338,339,340,341,342]. Familial cases of CCM exhibit autosomal dominant inheritance with incomplete penetrance and account for 10–50% of all cases. Disease-causing mutations have been identified in three genes: *KRIT1* (Krev interaction trapped 1) on 7q21-q22 (CCM1) [343,344,345,346,347,348], *MGC4067* (Malcavernin) on 7p13 (CCM2) [349,350,351], and *PDCD10* (Programmed cell death 10) on 3q26-q27 (CCM3) [349,352]. The proteins these genes encode for are involved in many angiogenic-related pathways and the maintenance of EC homeostasis, including apoptosis, cell-ECM adhesion, hypoxia response, and oxidative stress response [337]. However, it should be noted that both individuals with the inherited form of the disease and those without lesions can have an absence of germline and somatic mutations at the three CCM loci [336,337], pointing to further genetic factors contributing to the pathogenesis and progression of CCMs. Deep location (brainstem, cerebellar nuclei, thalamus, or basal ganglia) appears to be the most important predictor of subsequent hemorrhage of CCMs [353].

##### DAVFs

DAVFs are rare neurovascular malformations comprising of one or more arteriovenous shunts located in the dura mater [354]. They are acquired lesions, resulting from trauma and/or venous thrombosis. The hemorrhagic risk from DAVFs is estimated to be around 1.6–1.8% per year [355] and is primarily related to changes in hemodynamics when venous drainage is shifted from extracranial to intracranial routes. Venous hypertension is a hallmark of the disease process [354].

#### 6.2.3. Intracranial Aneurysms (IAs)

IAs are local dilatations in cerebral arteries that predominantly affect the circle of Willis. IAs present in approximately 2–5% of adults. The weakened areas of IAs are susceptible to rupture, leading to SAH [356]. Female sex, smoking, hypertension, and alcohol consumption are risk factors for IA, but a positive family history confers the greatest risk [357,358]. IAs occur mostly at bifurcations within the circle of Willis [359].

Structurally, there are two types of IAs: saccular and fusiform. Saccular (aka berry) aneurysms are sac-like pockets that arise from a cerebral wall. The less common fusiform aneurysms are dilations that affect a short length of the vessel where the entire vessel diameter is increased. IAs are one of the most common focal structural lesions that can cause hemorrhagic stroke [360]. The abnormal structure of IAs includes (1) torn, fragmented, or disappearance of internal elastic lamina [361], (2) rugged luminal surface of the intima [362], (3) myointimal hyperplasia [363], (4) disorganization of the muscular media [364,365], (5) hypocellularization [366], and (6) infiltration of inflammatory cells [367]. While typically associated with SAH, IAs may seldom present with isolated intraparenchymal hematomas, for example, deep hemorrhages with distal lenticulostriate aneurysms [368,369].

##### Conclusions and Future Perspectives

We provided a cogent discussion of the structure and function of the NVU and how it remodels after an ischemic and hemorrhagic stroke, as well as how it is predisposed by vascular risk factors from the perspective of cellular composition, gene expression profiles, signaling pathways and their physiological/functional manifestation. Since the conceptualization of the NVU, the vast amount of research has not only advanced our knowledge in how BBB contributes to neurological diseases including stroke and brain hemorrhage, but also revealed abundant targets of therapeutic intervention to minimize stroke injury and enhance recovery via preclinical models.

With evolving technology in structural and functional in vivo imaging and the development of novel molecular and transgenic tools, we will continue to gain an understanding in how each cell type and cellular component contributes to the integrity and regulation of the BBB in normal physiology and disease. The emerging engineered in vitro models of BBB containing ECs with various perivascular cell types in the presence of ECM building on microfluidic systems will also be instrumental in advancing our understanding in the mechanistic aspects of cell interactions in the NVU, providing a high-throughput system for the development of targeted therapies.

## Figures and Tables

**Figure 1 cells-11-02823-f001:**
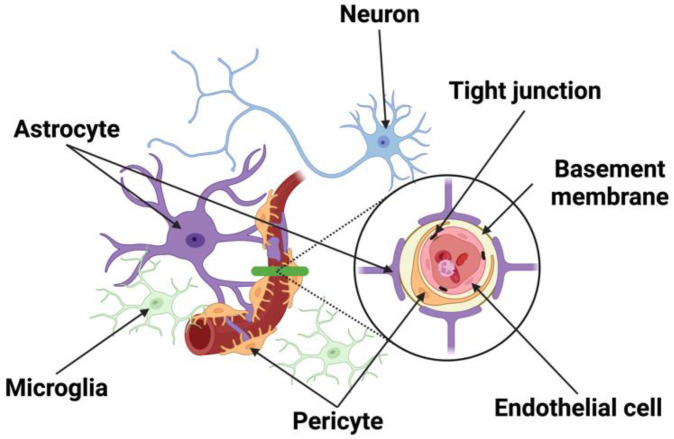
Schematic of the NVU. The NVU is comprised of neurons, vascular cells (endothelial cells, pericytes), basement membrane, and glia (astrocytes, microglia). Neurons make distinct connections with blood vessel and other cells of the NVU. Endothelial cells forming the blood vessels are encased by a basal lamina/basement membrane and are bound by tight junction proteins. Located in the brain parenchyma, astrocytes make contact with both pericytes and endothelial cells at the capillary wall, while pericytes are situated between the end feet of astrocytes and endothelial cells.

**Figure 2 cells-11-02823-f002:**
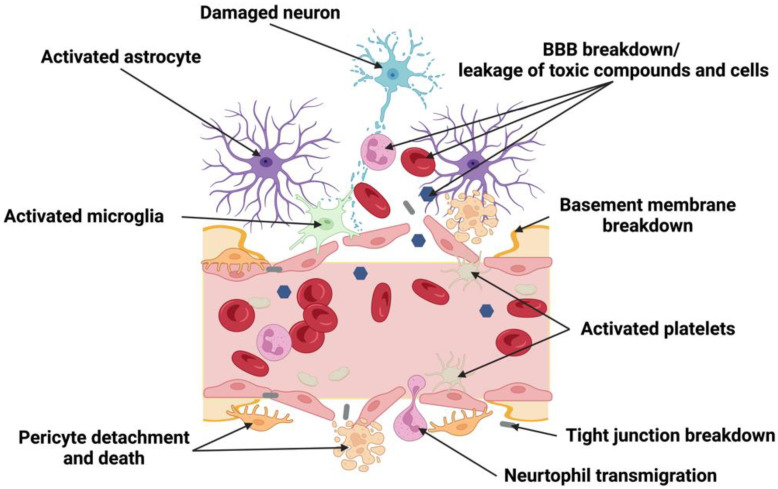
NVU dysfunction following ischemic stroke. A decrease of the cerebral perfusion leads to neuronal injury and death, causing an increase in the release of proinflammatory mediators that activates the components of NVU. Activated endothelial cells attract leukocytes, leading to their increased transmigration across the breached BBB and subsequent neuroinflammation. MMPs produced by activated endothelial cells and pericytes degrade the basement membrane, together with the tight junction breakdown they result in leakage of toxic compounds via BBB opening. Microglia dynamically transition into reactive and surrounding brain–blood vessels and exacerbate BBB breakdown, while activated astrocytes secrete proinflammatory mediators that cause further NVU disruption and neuronal injury. Furthermore, pericytes detach from vessel wall or die, further exacerbating BBB function and impair blood flow regulation.

**Table 1 cells-11-02823-t001:** Remodeling of NVU after ischemic stroke.

Endothelial cell	Following disruption of the BBB, the ferritin and the free iron accumulated in endothelial cells in brain capillaries enter the penumbra. Iron-dependent oxidative stress in the penumbra can lead to necrosis and further neurological deterioration following ischemic stroke [133].
	In an ischemia/reperfusion model of MCAO, BBB permeability exhibited a biphasic manner with permeability occurring at 3 and 72 h after reperfusion, and changes in claudin-5, occludin, and zonula occludens-1 protein levels [127].
	Fasudil hydrochloride recovered the neurological function, improved the function of BBB inhibited RhoA protein expression, and upregulated growth-associated protein-43 and claudin-5 protein expression following cerebral ischemia/reperfusion [134].
	Following a mild MCAO, tight junctions were stable during the first 24 h after reperfusion, but they underwent significant breakdown and remodeling from 48 to 58 h after reperfusion [128].
	4–24 h post-stroke, BBB breakdown was observed and vessels showed astrocyte end-foot swelling and increased endothelial vesicles [131].
Pericyte	2 h after hypoxia, pericytes started to migrate and one of every three pericytes migrated from the original location. In the first stage of migration, spikes occurred at the abluminal surface of pericytes [135].
	PDGFRβ expression was induced specifically in the pericytes in peri-infarct areas and its level was gradually increased [136]. In the cultured pericytes, PDGF-B induced cell growth and anti-apoptotic responses through Akt [136].
	Hypoxic-ischemic injury results in oxidative stress and pericyte constriction or eventual death [137].
	In photothrombotic stroke in mice, it was postulated that rapid activation of MMP-9 secreted from pericyte somata degraded underlying tight junction complexes and resulted in plasma leakage at places where pericyte somata adjoined the capillary wall [138].
	After pericyte ablation with diphtheria toxin, mice showed acute BBB breakdown, severe loss of blood flow, and a rapid neuron. Intracerebroventricular PTN infusions prevented neuron loss in pericyte-ablated mice despite persistent circulatory changes [139].
Astrocyte	Astrocytes are known to secrete pro-angiogenic factors that promote the growth of new capillaries toward the infarcted tissue [140].
	Reactive astrocytes have a protective role in brain ischemia, and the absence of astrocyte. IFs is linked to changes in glutamate transport, endothelin B receptor-mediated control of gap junctions, and plasminogen activator inhibitor-1 expression [141].
	After CNS injury, astrocytes induce angiogenesis by endogenous VEGF and upregulation of autocrine signaling, increasing both astrocyte proliferation and facilitating the expression of growth factors [142].
	CSDs induce vasoconstriction of vascular smooth muscle cells by increasing astrocytic vasoconstrictor 20-hydroxyeicosatetraenoic acid [143].
	Astrocytic gap junctions were implicated in propagating CSDs and leading to exacerbated brain damage [144].
	Astrocytes are activated by trophic factors produced by the pericytes within the ischemic regions, leading to peri-infarct astrogliosis [145].
Microglia and macrophage	Reactive microglia also secrete MMP-9 and MMP-3, proteases that can break down the basement membrane and exacerbate BBB leakage [146].
	Recruitment of microglia to blood vessels occurs within 6 h of reperfusion with significant accumulation in perilesional tissue. After 24 h of reperfusion, microglia fully enwrap small blood vessels in the peri-infarct region. Individual perivascular microglia displayed intracellular vesicles containing CD31-positive inclusions, suggesting phagocytosis of brain endothelial cells, which was correlated with BBB breakdown [147]. 72 h post-MCAO, blood vessel degradation was complete, and remaining vascular debris was cleared by microglia and immune cells
	P2RY12-mediated chemotaxis of microglia processes is required for the rapid closure of the BBB. Mice treated with the P2RY12 inhibitor clopidogrel, as well as those in which P2RY12 was genetically ablated, exhibited significantly diminished movement of microglial processes and failed to close laser-induced openings of the BBB. Thus, microglial cells played a previously unrecognized protective role in the maintenance of BBB integrity following cerebrovascular damage [148].

BBB: Blood–Brain Barrier, MCAO: Middle Cerebral Artery Occlusion, PDGF: Platelet-Derived Growth Factor, PDGFRβ: PDGF receptor β, MMP: Matrix Metalloproteinase, PTN: Pleiotrophin, IF: Intermediate Filament, CNS: Central Nerve System, VEGF: Vascular Endothelial Growth Factor CSD: Cortical Spreading Depression.

**Table 2 cells-11-02823-t002:** Predisposition of NVU to vascular risk factors.

Risk Factors	Findings
Diabetes and hyperglycemia	T1-weighted MRI showed a homogeneous high-intensity area in the corpus striatum in a diabetic hemichorea patient [194]
	Increased BBB permeability with MRI was detected in patients with type II diabetes or white matter hyperintensities [195]
	Immunostaining for 8-OHdG, a marker of oxidative DNA damage, was seen in the vessels of the cortex of 20-week-old diabetic mice [196]
	BBB permeability were significantly increased in diabetic rat, 5 weeks of rosuvastatin and simvastatin therapy (10 mg/kg) improved BBB permeability [197]
	Progressive increase in BBB permeability to small molecules from 28 to 90 days were observed in diabetic rat Insulin treatment attenuated BBB, especially during the first few weeks; however, as diabetes progressed, it was evident that microvascular damage occurred even when hyperglycemia was controlled [198]
	Diabetes increases BBB permeability via a loss of tight junction proteins, but increased BBB permeability in diabetes did not result from hyperglycemia alone [199]
	Hyperglycemia significantly compromised the BBB integrity and enhanced total PKC activity. Elevations in NADPH oxidase and MMP-2 activities and decreases in occludin levels contributed to barrier dysfunction [200]
	Upregulation of TJ-associated proteins were caused by insulin and idebenone in diabetic rats. The activations of ROS, AGEs, expression levels of RAGE and NF-κB were significantly decreased after insulin treatment [201]
	Hyperglycemia enhances amyloid precursor protein expression with increased in human umbilical vein endothelial cells. Amyloid beta-peptide production downregulates junctional proteins causing increased BBB permeability [202]
	Greater immunoreactivity of albumin was observed in the vessel wall of the periventricular area of diabetic mice [203]
	GLP-1 inhibited the increase in production of reactive oxygen species under hyperglycemia conditions and improved the BBB integrity induced by hyperglycemia [204]
	Significantly increased sodium fluorescein leakage in the hippocampus was observed in diabetic rat, and tight junction markers were significantly decreased in the hippocampi [205]
	BBB permeability in the diabetic monkeys was significantly increased [206]
	EX-4 improved the permeability of the BBB and the cognitive parameters in diabetic rat [207]
	The astrocyte foot processes were detached, and the microglial cells played an invasive damaging role in diabetic mice. Endothelial cell deterioration was observed: loss of electron density, basement membrane thickening and rearrangement, increased transcytotic-pinocytotic vesicles, and aberrant mitochondria [208]
	Pericyte apoptosis was detected in vivo in hyperglycemia rats and in vitro in retinal cultures [209]
Hypertension	Cerebral arterioles underwent remodeling and hypertrophy in transgenically hypertensive mice [210]
	The staining for HRP was distributed around the vessels in the hippocampal fissure of spontaneous hypertensive rats [211]
	Increased BBB permeability and impaired cognitive functions was observed in angiotensin II-infused hypertensive mice; Angiotensin II receptor blocker markedly ameliorated leakage from brain microvessels and restored the cognitive decline [212]
	Hypertension significantly decreased the brain glutathione content and increased the brain malondialdehyde level [213]. The mRNA levels of claudins (3, 5, and 12) proteins significantly decreased in response to hypertension;
	Tight junction was destroyed gradually 8 weeks after hypertension, and the levels of zonula occludens-1 and occludin also decreased gradually [214]
	In SHR brain, an obvious glial reaction was found both for GFAP-immunoreactive astrocytes and for microglia; The pro-inflammatory IL-1β was significantly increased in CA1 sub-field of SHR hippocampus and the TNFα expression was higher in frontal cortex of SHR compared with WKY [215]
	Chronic hypertension significantly increased spontaneous Ca^2+^ events within astrocyte microdomains [216]
Hyperlipidemia	70% more spontaneous leakage of injected Evans blue dye in the brains of APOE knock-out mice [217]
	BBB permeability was higher in APOE4 knock-in mice than in APOE3 knock-in mice, suggesting that TJ integrity in BBB is regulated by APOE in an isoform-dependent manner [218]
	The expression of APOE4 and lack of murine APOE led to BBB breakdown [97]
	Increase of mean Ki during the first 20 min after infusion of human TGRL lipolysis product was observed in mice [219]
Aging	The choline carrier in old rats had reduced capacity and increased affinity [220]
	Aged mice showed significant attenuation in the expression of BBB tight junction proteins. TNF-α in cerebral endothelial cells of aged mice was elevated and this was associated with heightened peripheral inflammation [221]
	Age-dependent BBB breakdown in the hippocampus was observed. The BBB breakdown in the hippocampus and its CA1 and dentate gyrus subdivisions worsened with mild cognitive impairment [222]
Sex	2- to 4-fold increase in dye extravasation in the olfactory bulb and hippocampus of reproductively senescent female mice; estrogen reduced dye extravasation in young adults compared with age-matched counterparts [223]
	Ovariectomy induced a 2.2-fold increase in Evan’s blue dye extravasation into the brain in young female mice. The expression of the tight junction protein in microvessels was not altered [224]
	Treatment of endothelial cells with 17beta-estradiol led to an increase in transendothelial electric resistance and a most prominent upregulation of the tight junction protein claudin-5 expression [225]. A significant increase of claudin-5 promoter activity, mRNA, and protein levels was also detected
	Estradiol prevented inflammation-induced defects in barrier function in mice [226]

MRI: Magnetic Resonance Imaging, BBB: Blood–Brain Barrier, 8-OHdG: 8-hydroxy-2’-deoxyguanosine, PKC: protein kinase C, NADPH: Reduced nicotinamide adenine dinucleotide phosphate MMP-2: matrix metalloproteinases-2, TJ: Tight Junction, ROS: Reactive oxygen species, AGEs: advanced glycation end products, RAGE: receptors for advanced glycation end-products NF-kB: nuclear factor-κB, GLP-1: glucagon-like peptide-1, EX-4: exendin-4, HRP: horseradish peroxidase, SHR: Spontaneously hypertensive rats, GFAP: glial fibrillary acidic protein, IL: interleukins TNFα: tumor necrosis factor alpha, WKY: Wistar Kyoto rats, APOE: Apolipoprotein E, TGRL: triglyceride-rich lipoprotein.

## Data Availability

Not applicable.

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
