# Peer review of "Remodeling of the Neurovascular Unit Following Cerebral Ischemia and Hemorrhage"

_cells, 2022, doi:10.3390/cells11182823_

Round 1

Reviewer 1 Report

The review summarizes main knowledge about neurovascular unit (NVU) cell components and their role in both physiological and pathological conditions. In detail, it focuses on vascular damage and remodeling, following stroke. The review is globally and well written. However, there are some aspects to consider before publication.

Main revisions:

- the first paragraphs are very redundant and repetitive, making reading boring. At the same way, 4. and 5. subsections.

- Please include some images. There is a very huge text amount

- In the AVM section, please note that there are two recent papers that describe germline mutations and related enriched pathways in bAVM patients (PMID: 35228337, PMID: 32560555) that can contribute to increased BBB permeability.

- In the CCM section, please note that there are two important expression studies describing dysregulated pathways in CCM endothelial cells (PMID: 35771000PMID: 32877751).

Minor revisions:

- Correct line 242

- Correct line 274

Author Response

The review summarizes main knowledge about neurovascular unit (NVU) cell components and their role in both physiological and pathological conditions. In detail, it focuses on vascular damage and remodeling, following stroke. The review is globally and well written. However, there are some aspects to consider before publication.

Main revisions:

- the first paragraphs are very redundant and repetitive, making reading boring. At the same way, 4. and 5. subsections.

Response: We apologize for the overlapping information from various aspects of the discussion. We have since revised the ms and streamlined the discussion and eliminate the redundancy. We just want to point out that deletion of the text, English editing or re-structuring is wide-spread and not tracked, while addition of text is highlighted in red text.

- Please include some images. There is a very huge text amount

Response: We now included 2 figures to illustrate the make-up of the NVU and the breakdown caused by ischemia.

- In the AVM section, please note that there are two recent papers that describe germline mutations and related enriched pathways in bAVM patients (PMID: 35228337, PMID: 32560555) that can contribute to increased BBB permeability.

- In the CCM section, please note that there are two important expression studies describing dysregulated pathways in CCM endothelial cells (PMID: 35771000 , PMID: 32877751).

Response: We are grateful for this valuable information and since incorporated the essence of these findings as highlighted in red tex.

Minor revisions:

- Correct line 242

- Correct line 274

Response: Both typos corrected.

Reviewer 2 Report

The authors prepared a comprehensive review article on the neurovascular unit (NVU) and the remodeling that occurs after cerebral ischemia and hemorrhage. In the text they first provide an explanation of the blood brain barrier and the elements that regulate brain homeostasis. The information provided spans from the structural elements that make up the NVU to the biological responses that occur because of ischemic and hemorrhagic stroke, and it is organized in a digestible manner into sections based on cell type. Below are some suggestions for the authors

 1.     The formatting of the table needs to be changed, perhaps listing information in the form of bullet points.

2.     A schematic diagram or a graphical abstract depicting the interaction of NVU elements during ischemia or hemorrhage (also one that compares the similarities and differences between ischemia and hemorrhage in the context of NVU remodeling) would be a welcome addition to this paper.

Minor:

  • Line 119: “Voltage-dated” should be corrected to “Voltage-gate”
  • Line 241-242: First sentence is missing a period
  • Line 445: Heading should read “3.3” and not “2.3”

Author Response

  1. The formatting of the table needs to be changed, perhaps listing information in the form of bullet points.

Response: We appreciate this great suggestion and covert the table to bullet point format.

  1. A schematic diagram or a graphical abstract depicting the interaction of NVU elements during ischemia or hemorrhage (also one that compares the similarities and differences between ischemia and hemorrhage in the context of NVU remodeling) would be a welcome addition to this paper.

Response: We now included 2 figures to illustrate the make-up of the NVU and the breakdown caused by ischemia.

Minor:

  • Line 119: “Voltage-dated” should be corrected to “Voltage-gate”
  • Line 241-242: First sentence is missing a period
  • Line 445: Heading should read “3.3” and not “2.3”

Response: Typos corrected.